# Numerical Investigation of 1 × 7 Steel Wire Strand under Fretting Fatigue Condition

**DOI:** 10.3390/ma12213463

**Published:** 2019-10-23

**Authors:** Sajjad Ahmad, Saeed Badshah, Ihsan Ul Haq, Suheel Abdullah Malik, Muhammad Amjad, Mohd Nasir Tamin

**Affiliations:** 1Department of Mechanical Engineering, International Islamic University, Islamabad 44000, Pakistan; sa.ahmad@iiu.edu.pk (S.A.); m.amjad@iiu.edu.pk (M.A.); 2Department of Electrical Engineering, International Islamic University, Islamabad 44000, Pakistan; ihsanulhaq@iiu.edu.pk (I.U.H.); suheel.abdullah@iiu.edu.pk (S.A.M.); 3Faculty of Mechanical Engineering, Universiti Teknologi Malaysia, Johor Bahru 81310, Johor, Malaysia; taminmn@fkm.utm.my

**Keywords:** finite element analysis, wire rope/strand, fretting fatigue, axial loading

## Abstract

Wire ropes undergo a fretting fatigue condition when subjected to axial and bending loads. The fretting behavior of wires are classified as line contact and trellis point of contact. The experimental study on the fatigue of wire ropes indicates that most of the failure occurs due to high localized stresses at trellis point of contact. A continuum damage mechanics approach was previously proposed to estimate the fatigue life estimation of wire ropes. The approach majorly depends on the high value of localized stresses as well as the micro-slippage occurs at the contact region. Finite element approach has been used to study radial and axial distribution of stresses and displacement in order to clearly understand the evolution of stresses and existence of relative displacements between neighboring wires under various loading and frictional conditions. The relative movements of contacting wires are more when friction is not considered. In the presence of friction, the relative movement occurs at the boundaries of the contact region. The location of microslip in the presence of friction is backed by the experimental observation stating the crack is initiated at or the outer boundary of the contact spot. The existence of slip is due to different displacement of outer and central wires.

## 1. Introduction

Steel wire ropes are used in many engineering applications like mooring lines of FPSO and turbines, aerial rope ways, ski lifts, mine hoisting and lifting cranes [1,2]. They have capabilities to withstand high axial load along with comparatively small bending and torsional loads. A wire rope consists of several individual wires having different types of contacts with each other. Two classes of interwire contact occur in a typical multi-layered spiral strand [3]. The first, which influences the overall axial, torsional, and free bending stiffnesses, and governs the associated hysteresis values under cyclic loading, is the contact within a given layer between adjacent ‘parallel’ wires. This is called the line contact. This type of contact will influence not only the elastic properties but also the fatigue life. Any fretting on the line contact will significantly reduce the fatigue life from that of the constituent wires which is controlled by their surface finish and residual stresses. The other class of interwire contacts often occurs between the layers of a spiral strand and is called the point contact or trellis contact. Because they are localized, the contact stresses are much higher than on the line contacts within a layer. These localized stresses affect the elastic, hysteretic and fatigue properties of wire ropes. These contact points have received more attention by researchers [4,5,6,7] due to their major contribution in the failure of wire ropes. In many designs of wire ropes, this trellis contact is minimized by using some lubricants between layers of wires. It has been observed that fracture occurs in these trellis contacts during fatigue testing. Since the amplitude of fretting movement is small, partial slip occurs and the crack is generated at or toward the edge of contact spots between wires. When the amplitude becomes larger than fretting, wear and failure may occur due to a reduced cross-section and increased stresses. Design against failure under tensile loading exceeding the MBL (maximum breaking load) is already established [8]. Fretting fatigue life was estimated from the fretting fatigue data of individual wires along with Finite Element analysis of wire ropes [9] using a stochastic approach. Fatigue design of a wire rope under high cycle fatigue under fretting conditions have been investigated [2,10,11,12]. The data of fatigue life are based on available standard DNV OS-E301 (2004) and published literature by researchers [13,14]. A continuum damage mechanics approach was proposed recently by [15] for estimating fatigue life of wire rope. This approach is based on the Lemaitre model [16] for high cycle fatigue. This is a two-scale model considering damage as micro-inclusion embedded in elastic matrix. The drawn steel of wire rope under high cycle fatigue loading behave in quasi-brittle manner where the behavior is brittle at the meso level and damage occurs at the microlevel. The damage starts to occur from the trellis point of contact region where the contact stresses are high and micro slippage occurs at the mating surfaces. In literature, there are not enough FE (Finite Element) studies which describe the contact behavior of wires. D. Wang et al. [2] did stress analysis of three-layered wire strands. The analysis was carried out for 1/6th pitch length of wire rope producing stress distribution as well as relative displacement between neighboring wires. In addition to these analyses, Archard’s wear law was used to predict depth of wear for different wires crossed at different angles. The results were focused on fretting wear which occurred due to line contact between wires and which do not consider the trellis point of contact between wires. Yu Yujie et al. [17] did an FE-based study on 1 × 7 strands under longitudinal as well as lateral loads. The distribution of stresses and strains were studied in both conditions and compared with each other. The wire strand model had outer wires touching each other making line contact with each other in the beginning of loading and then, due to Poisson effect, it reduced and the stress due to contact between core and outer wires in the lateral part of loading increased. The sliding and frictional energies for longitudinal loading were described and analyzed for the contacting nodes. Goran Vukelic and Goran Vizentin [18] used FE for studying possible remaining service life of wire ropes. A single wire or strand is suppressed or deleted after certain cycles and von Mises stresses were found out for the reduced cross-section of the wire rope. Using the Goodman approach for finite fatigue, possible remaining service life was studied.

This paper describes the evolution of stresses at the localized region, radial and axial distribution of the stresses and displacement at the neighboring surfaces of core and outer wires under different contact conditions. The outer wires have no contact with each other; hence, the mechanism of contact will be different compared with early FE studies. These studies were comparatively less focused on the concentrated stresses due to spot contact between wires. Friction between the contacting wires is another important factor effecting the fretting fatigue. Friction is unavoidable due to environmental (sea and air etc.) effects, although lubricant and surface treatment is done [7] to minimize the friction. The role of lubricants and grease existed in the initial cycles, but it faded away in the later life due to high contact pressures between wires. This study will probably be helpful in understanding the evolution of contact stresses, strains and interaction of individual wires (fretting) under axial loading of one cycle having different surface conditions.

## 2. Materials and Methods

### 2.1. Geometry of Wire Rope

The wire rope strand 1 × 7 consists of one central and six outer wires. All the wires were made of cold drawn steel. Wire rope strand is modeled using Finite Element commercial software ABAQUS (6.12, Dassault Systemes, Johor, Malaysia). The central wire was modeled using a solid extrusion method and an outer wire was modeled with same extrusion method, but information of the pitch length was given. The pitch length is important in defining the lay angle of outer wires around the central wire. The pitch length of the existing strand was 230 mm, but the full length was 330 mm. An additional length was provided to eliminate the end effects [6,19,20]. End effects were significant in the strand terminals due to boundary condition which were not under consideration in the present study. Length of any strand used for any application will be the multiple of many pitch lengths, so a pitch length is considered as a unit of full length of strand. The interaction of wires under certain axial loading will be nearly the same in any pitch length of the wire strand. Central and outer wires were assembled in the Assembly module. The central wire diameter was 5.43 mm and outer wire diameters are 5.23 mm.

### 2.2. Material Properties and Structural Mesh

All the wires of 1 × 7 strands were considered as isotropic and were homogenously made of drawn steel. The chemical composition of the drawn wires was (in wt.%) 0.83C, 0.91Si, 0.717Mn, 0.0124P, 0.0031S, 0.015Cu, the remaining being Fe [15]. The modulus of elasticity was 202 GPa and Poisson’s ratio is 0.28 [15,21]. In addition to the elastic information, stress–strain information after yielding was also provided [21] and the wire strand was subjected to loading below 50% MBL but as the stresses were more in the trellis point of contact region [5,6], plasticity could not be ignored. A hexahedral mesh element C3D8R was employed for the discretization of the entire strand with a total number of 128260 elements and 160710 nodes shown in Figure 1. Each node has three degrees of freedom i.e., U1, U2, U3 (translation in X, Y and Z directions).

### 2.3. Interaction Properties and Boundary Conditions

The interaction of wires was defined in the interaction module of ABAQUS. The contact pairs were automatically identified, and a master–slave algorithm was employed between the contact pairs with finite sliding slip mode enforced at the contact constraints. For the frictionless contact cases, only normal contact (hard) behavior was defined but for the friction cases the tangential contact properties were defined by applying penalty algorithm with a coefficient of friction of 0.1 and 0.2. the values of coefficient of friction are selected to show the effects on the local stresses and relative displacements at the contact due to existence of friction. Two reference points “Fixed End” and “Loading End” shown in Figure 1 were defined along centerline of central wire away from the strand cross section on both the front and back side. Kinematic coupling was employed between these reference points and the surface nodes on the cross section of strands. The coupling nodes are constrained with all degree of freedom. One reference point (Fixed end) is fixed by restricting all degrees of freedom and load is applied at other reference point (Loading end) in U3 (Z axis) direction. Loading is applied in three steps. In first step ramp loading is applied upto minimum load, then single cycle is applied keeping the R ratio equal to 0.1. Maximum load value of 80 kN, 120 kN and 145 kN is applied.

## 3. Results

### 3.1. Stress Distribution under Different Amplitude of Cyclic Loading

When cycling load was applied on wire rope strand with R = 0.1, stresses ranged from minimum to maximum values. Stresses in the central wire were higher as compared to outer wires due to difference in length. The wires which are helically wrapped around are higher in length and result in less value of stress and larger value of displacement. The average value of stress in any layer of wire rope strand can be found mathematically according to [22],
(1)$$\sigma_{\mathrm{tk}}=\frac{\frac{\cos^{2}\alpha_{k}}{1+\upsilon_{k}\sin^{2}\alpha_{k}}E_{k}}{\sum_{i=0}^{n}\left(\frac{z_{i}\cos^{3}\alpha_{i}}{1+\upsilon_{i}\sin^{2}\alpha_{i}}E_{i}A_{i}\right)}S$$
where index i stands for the individual wire and k for the layers; A is the cross-sectional area of the wire, E is the modulus of elasticity, υ is Poisson’s ratio, k is the lay angle, z is the number of wires in the layer, n is the number of wire layers and S is the tensile force.

As the drawn steel wire is not flexible enough to wrap completely around the central wire, there were certain points and regions which came under spot contact. This contact is not uniform and continuous due which stresses are comparatively high in some regions [4,7] and result in localized stress concentrations. This kind of contact was called the trellis point of contact, having a significant role under fretting fatigue conditions. The trellis point of contact was observed in different kinds of rope design and under different types of loadings (axial, bending, torsion and mix loading). Figure 2 shows the distribution of von Mises stresses at 80 kN load extracted at peak load occurring at the central section of a strand. Sections from the central region of strand were selected to avoid the terminal effects due to boundary conditions. The central wire was comparatively high stressed as compare to surrounding wires due to difference in length of the wires. The stresses in the central wire were not homogenously distributed due to high localized stresses in some regions. Radial distribution of von Mises stresses is shown in Figure 3a for all the three loadings and frictionless and friction cases. Results were extracted at the peak value of loading for the path defined at cross sectional area in the middle region of wire rope. This path starts from the center of core wire (CW) towards the outer wire (OW) passing through the contact point of the two wires. The trends are almost the same in all cases with maximum values at the surface of the central wire having a radius of 2715 mm; they then suddenly jumped down to certain lower values and then gradually decreased. More detailed insight of theses stresses at the contact region can be seen in Figure 3b,c which are plot against the true distance along the strand. Results are extracted for one pitch length of strand chopping 50 mm sections at each end. These results were extracted at the peak load of one cycle.

Von Mises stresses are maximum in the central wire; the neighboring wires in both the frictional and frictionless cases as indicated in Figure 3. In the frictionless case the von Mises, stress is more uneven on the surface path. There are some randomly occurring peaks showing trellis localized stresses. In case of friction, the overall trend is constant with randomly occurring peaks, showing localized stresses. The friction has very low impact on the stress distribution of neighboring wires as indicated. The trend and distribution of peaks are almost the same as in the frictionless cases for all three types of loading. Results having friction coefficient values of 0.1 and 0.2 are almost the same with no significant differences. It is important to note that none of the values of von Mises stress exceed the values of yield stress [15] of the drawn steel wire. It means that loading below 50% MBL will cause no plasticity in the localized region. Considering the High Cycle Fatigue in which metals fails like quasi-brittle material, a ductile may undergo brittle failure if the material is subjected to low level of stresses causing elastic behavior at the meso-level. At the micro level, plasticity may occur [15,16] and will initiate damage in the subsequent cycle. In the absence of friction, neighboring wires tend to slip with each other which will be discussed in the following section. The friction which may be due environmental effects or other reasons (during manufacturing) resists the slippage of neighboring wires. If the frictional effects are increasing, the phenomena will change from fretting fatigue to fretting wear.

### 3.2. Relative Displacement of Central and Outer Wires

In the previous section the effect of different magnitude of loadings and surface roughness on distribution of von Mises stresses were discussed. It was revealed that results for frictionless case were different from the friction cases. Further increasing the friction from 0.1 to 0.2 did not cause significant effects on the results. Outer wires (OW) are helically wrapped around the central wire due to which stresses are not uniformly distributed across the cross-section of a strand. Fretting regions were further investigated by looking at the results of displacements. The contours of total displacement (U, magnitude) and axial displacements (U3) are shown in Figure 4 for 80 kN load. Contours of total displacement show variations in values for central and outer wires; however, no such variation has been noted for distribution of axial displacement. Since the total displacement shows some variations, it is further illustrated using radial distribution of total displacement for 80, 120 and 145 kN, as shown in Figure 5a. The radial path defined for this purpose is the same as illustrated in Section 3.1. The displacement of wire nodes is minimal at the center and increases radially. In the presence of friction, the displacement is continuous due to the sticking effect of the outer wire with a central wire. A sharp discontinuity can be seen in the frictionless cases at the boundary of neighboring wires, which is the result of relative slipping of mutually contacted wires.

Looking into the details of this relative slipping, the value of total displacements are drawn on the axial path at the contact of the wires as shown in Figure 5. Figure 5b illustrates the displacement of core and outer wires (OW) in the absence of friction. The displacement curves are not exactly overlapping, hence, showing some relative displacement of both the core and outer wires. It is worth mentioning that the axial displacement, as shown in Figure 5c, illustrates no significance difference between the core and outer wires. This shows that the relative displacement may be due to the untwisting effect of strands when axial force is applied at both ends. In case of friction, for the 80 kN load case, the displacement of the outer wires is more than the core wire, but at larger loads, this difference further decreases as illustrated in Figure 5c. Due to friction at the contact region slipping and being resisted, both the neighboring wires stick to each other. This slipping and sticking is shown in Figure 6. In case of frictionless contact, there is no sticking which means no shear traction, but in case of friction, sticking occurs at all the loads. The slipping region reduces and shrinks at the boundaries of the contact region which means that both sticking and slipping occurs when the surfaces are rough enough to induce friction. This finding is according to previous experimental investigation [6,7,23]. The slipping (green) is less, as can be seen in Figure 6b, as compared to sticking (red) region. Figure 6b reveals the fact that in case of friction, partial slipping occurs at the contact region and the existence of fretting cannot be eliminated.

### 3.3. Stresses at Contact Points

In the preceding sections, radial and axial distribution of von Mises stresses were discussed along with distribution of axial displacement and total displacements of wires for all cases. In the current section, evolution of stresses at the contact points are discussed. The contact stresses and von Mises stresses were plotted for the adjacent nodes of the central core and outer wires. These nodes are shown in Figure 7.

These nodes are taken as samples from the central region of the wire rope strand to eliminate the end effects on the strand. The other spots where contact exists can be clearly observed in these Figures. The central core wire (CW) and outer wires (OW) are chopped near these nodes to show their exact location. The evolution of contact stresses CPRESS are shown in Figure 8 for 80 kN maximum load for both frictionless and friction cases. The evolution of the contact stress at a localized region in the frictionless case is more uneven as compared to the frictional case. The shape of contact stress does not exactly follow the shape of applied loading due to gross slipping behavior of the adjacent nodes. The load with R ratio of 0.1 is maintained on the strand ends. In the frictional cases, both the adjacent nodes have an exact shape due to sticking behavior. Fretting does exist in the frictional cases as previously discussed at the boundaries of contact areas. The distribution of von Mises stress at the core nodes is also shown for illustration purposes in Figure 8b.

The shape of von Mises stresses resembles the applied loading for all loading cases. The frictionless case for 80 kN load as shown in Figure 8 and is more fluctuating as compared to the frictional cases, clearly showing the gross relative slipping behavior of the nodes. The sticking behaviors in the frictional cases make it a smoother curve. At larger loads of 120 kN, von Mises stresses are relatively smooth, having a similar shape and trend. The peak value of von Mises is larger for the frictionless case as compared to the frictional case. This behavior shows that the magnitude of the applied load affect the evolution of stresses. Further investigating the von Mises stress at 145 kN load, the shape of the distribution is the same but not very well continuous. The small fluctuation in the stresses may be due to the increase in the slipping behavior at larger loads. All these results show that slipping behavior changes when the amount of loading and the condition of the mating surface change.

## 4. Discussion

Stress distribution among the wires of 1 × 7 strands, its displacement distribution and contact stresses were studied in this paper. Stress distribution in the wires validate the analytical Equation (1) [22] showing high values of stress in the core wire as compared to outer wires. These stress distributions were shown in Figure 2 and Figure 3. The peaks in the axial distribution of von Mises stress were due to localized stresses. These localized stresses were previously observed in experimental work by [23,24] and were considered as the reason of failure of wire ropes. It was observed in these studies that fretting fatigue occurs at the interlayer contact spots in multilayer wire ropes. However, in our present work, a wire rope strand is used (with only one layer). The confirmation of the fretting phenomenon was the primary focus in this study before implementing the methodology proposed by Ahmad et al. [15]. As indicated earlier, the outer wires do not contact each other, resulting in high value of stress at different contact points between the core and outer wires. Earlier FE studies [2,17] ignored these contact spots and primarily focused on line contact between wires resulting in a gross slip between wires confirming the fretting wear phenomenon. The other parameter which is necessary for the fretting phenomenon is the relative movement between contacting wires. This paper describes that slippage is maximum for the frictionless case and decreases when friction increases as indicated in Section 3.2. The relative displacement in the axial direction is negligible but the resultant vale of displacement shows some differences indicating some twisting behavior of helical wires. The contact spots are clearly shown in contours of CPRESS resembling the contact spots shown in morphologies reported by [7]. 

Morphologies of damage portions of a wire reported by [7] clearly validate the FE study showing the contact spots. The cracks are initiated at the boundaries of contact spots due to micro-slippage occurring between neighboring wires. Cracks are a confirmation of the fretting phenomenon contrary to wear in which material is removed at the contact spot resulting in failure of wire due to reduced cross-sections. At the beginning of loading friction, the value of wires is usually less [25] but increases with the passage of cycles deteriorating the surface. In the high cycle of fatigue amplitude, loading is very small and results in very small relative movement between wires. The values of maximum von Mises stresses were sinusoidally increased with the passage of an increase in loading. Maximum values are noted at the peak of loading cycles at these contact spots for different loads. It has been observed that von Mises stress values are below the yield stress at 50% MBL validating the domain of high cycle fatigue.

## 5. Conclusions

It was concluded from the numerical investigation of wire rope strands, that at low levels (below 50 Percent MBL) of stress, yielding does not occur at any point in the strand. The maximum value of the von Mises stress is 1554 MPa which is 91 Percent of the yield strength. Elastic behavior of the material shows that the HCF study of the wire rope strands is valid at these values of stresses.The shape of the von Mises stresses and the contact stress evolution during loading of one cycle are in phase with the applied loading at the localized contact regions.Relative displacement between the contacting wires is more in frictionless conditions. For friction, case relative displacement has been observed at lower loads of 80kN, however, changing the friction coefficient from 0.1 to 0.2 did not alter the results.It has been evident from the present study that fluctuation in stresses are more for 80 kN and 145 kN as compared to 120 kN for mutually contacted nodes. This fluctuation in stress changes is due to the sliding behavior of neighboring wires.At larger axial loads, the normal contact stress values are more for the frictionless case as compared to friction.The existence of the slippage at the boundaries of contacting spots in the presence of friction confirms the fretting phenomena backed by previous experimental studies stating that the cracks tend to nucleate at a spot or at the outer edge of contact spots.

## Figures and Tables

**Figure 1 materials-12-03463-f001:**
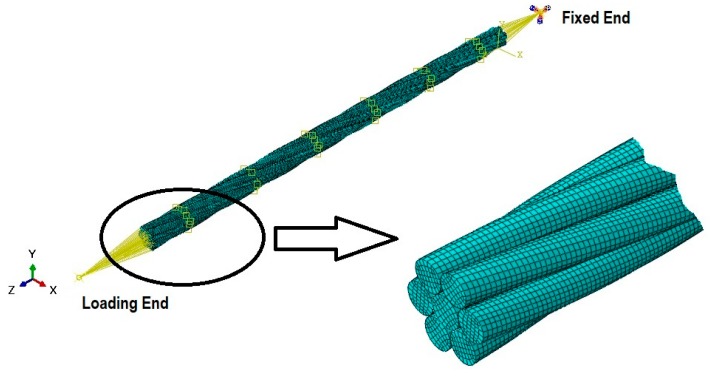
Wire Rope Strand Interaction, Boundary Conditions and detail of Mesh.

**Figure 2 materials-12-03463-f002:**
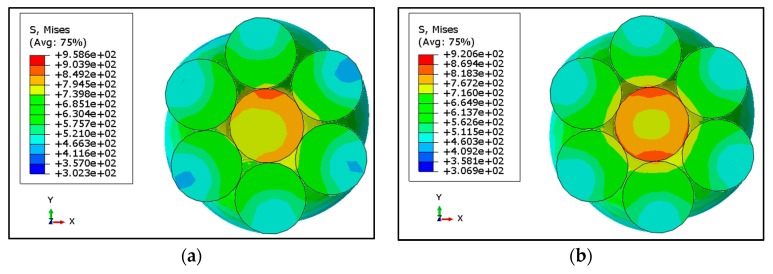
Von Mises and Max Principal stress distribution in wire rope strand under axial loading for different surface conditions (**a**) Von Mises stress distribution without friction. (**b**) Von Mises stress distribution with coefficient of friction = 0.1. (**c**) Von Mises stress distribution with coefficient of friction = 0.2. (**d**) Max Principal stress distribution without friction. (**e**) Max Principal stress distribution with coefficient of friction = 0.1. (**f**) Max Principal stress distribution with coefficient of friction = 0.2.

**Figure 3 materials-12-03463-f003:**
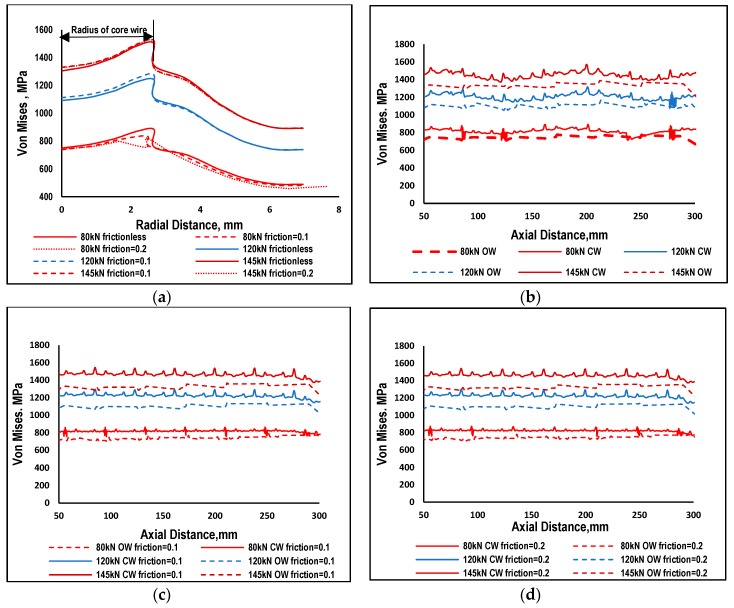
(**a**) Radial Distribution of von Mises Stress at mid-section of wire rope subjected to different axial loads (**b**) Von Mises along the axis of rope on the surface of core wire (CW) and outer wire (OW) in absence of friction. (**c**) Von Mises along the axis of rope on the surface of core wire (CW) and outer wire (OW) in presence of friction with coefficient of friction = 0.1. (**d**) Von Mises along the axis of rope on the surface of core wire (CW) and outer wire (OW) in presence of friction with coefficient of friction = 0.2.

**Figure 4 materials-12-03463-f004:**
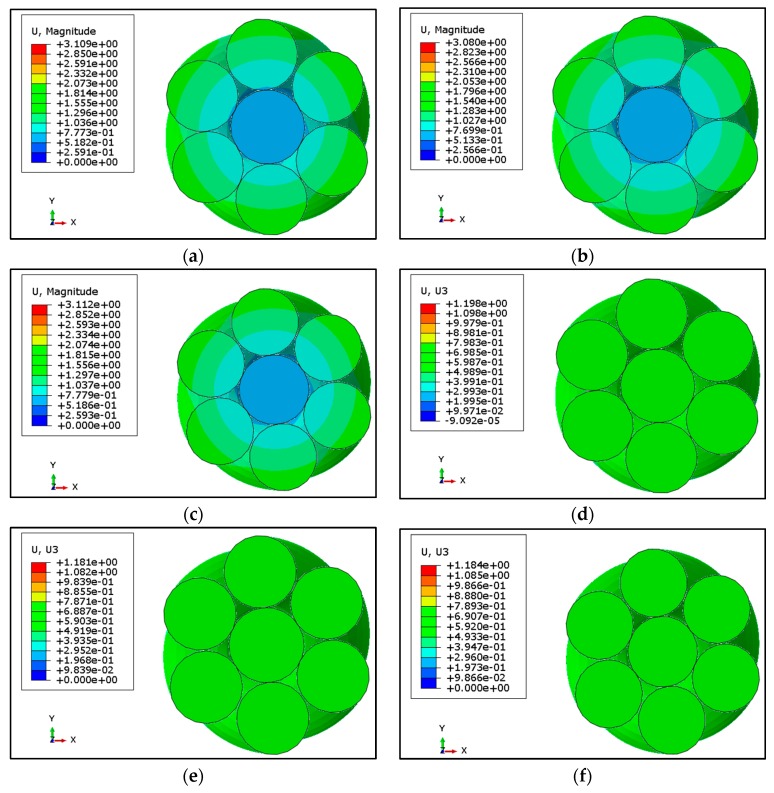
Total displacement (U, Magnitude) and Axial Displacement U3 in wire rope strand under axial loading for different surface conditions (**a**) Total Displacement without friction. (**b**) Total Displacement with coefficient of friction = 0.1. (**c**) Total Displacement with coefficient of friction = 0.2. (**d**) Axial Displacement (U3) without friction. (**e**) Axial Displacement U3 with coefficient of friction = 0.1. (**f**) Axial Displacement with coefficient of friction = 0.2.

**Figure 5 materials-12-03463-f005:**
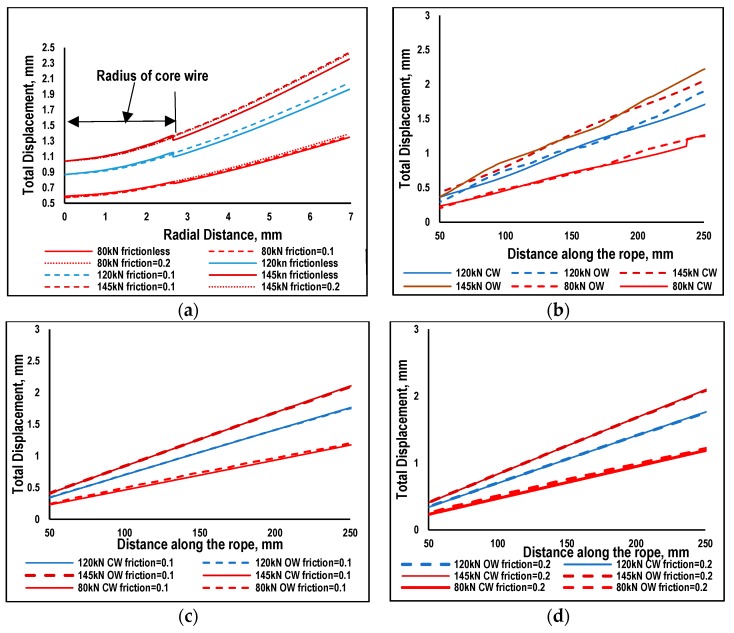
(**a**) Variation in Total Displacement along radial path at mid section of wire rope under different axial load and different surface conditions (**b**) Variation in Total Displacement in core and outer wire (OW)along the axis of wire rope under different axial loads in absence of friction (**c**) Variation in Total Displacement in core (CW) and outer wire (OW) along the axis of wire rope under different axial loads having coefficient of friction = 0.1. (**d**) Variation in Total Displacement in core (CW) and outer wire (OW) along the axis of wire rope under different axial loads having coefficient of friction = 0.2.

**Figure 6 materials-12-03463-f006:**
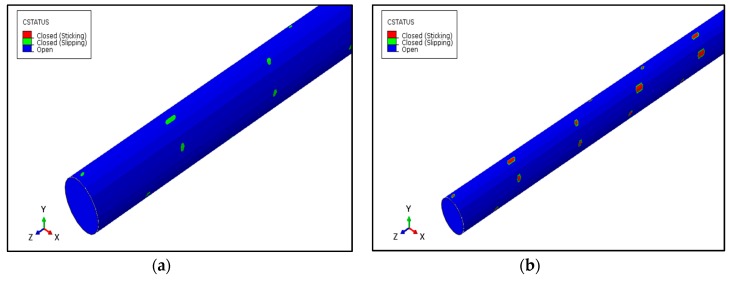
(**a**) Spots on Core Wire showing slipping with neighboring wires when there is no friction. (**b**) Spots on Core Wire showing sticking and slipping with neighboring wires when friction is present.

**Figure 7 materials-12-03463-f007:**
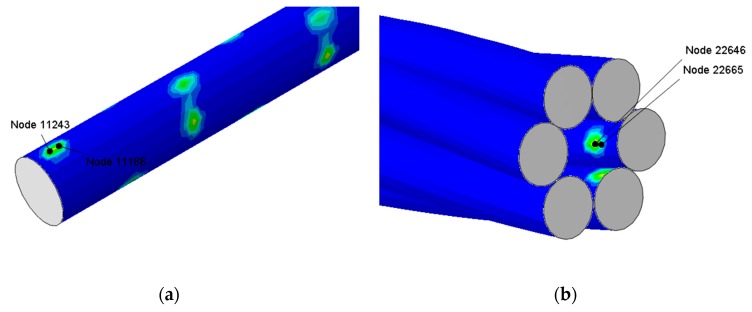
(**a**) Contact Nodes on Core Wire (**b**) Contact Nodes on Outer Wire.

**Figure 8 materials-12-03463-f008:**
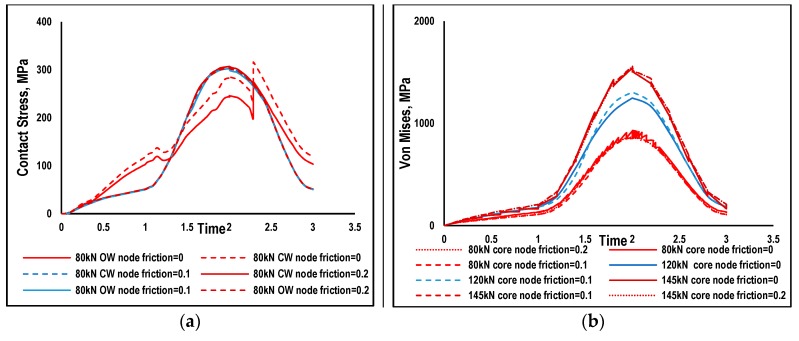
(**a**) Contact Stress at Neighboring nodes (**b**) Von Mises Stress at Neighboring nodes under different axial loads.

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
