# Peer review of "Numerical Investigation of 1 × 7 Steel Wire Strand under Fretting Fatigue Condition"

_materials, 2019, doi:10.3390/ma12213463_

Round 1
Reviewer 1 Report
The authors present a finite element analysis of the stresses in a twisted steel wire rope, focusing on the effect of different friction conditions on the contact between the individual strands.
Overall the work is sound, but in certain areas the presentation could be improved to make it easier for the reader to understand the paper. Further justification of two of the conclusions is also necessary. Specific comments follow below:
1) Conclusion 3 states: 'Relative displacement is more when there is less friction between the contacting wires'. I agree that Figure 6 shows a difference between the frictionless, and with friction, conditions, but no difference is shown between the two friction coefficients. Perhaps it would be better to state: There is more relative displacement in the frictionless condition? Or the authors should explicitly show that there is more relative displacement for a friction coefficient of 0.1 than with a coefficient of 0.2
In addition, explicitly showing the relative displacement (e.g. in Figure 6, or in a new figure) would also be helpful to justify this conclusion.
2) It is not clear to me exactly what the authors mean with conclusion 4. It is also not clear to me where it is shown that the 'fluctuation in stresses are more for 80 kN and 145 kN as compared to 120 kN'
3) Figure 1: the yellow text on a white background is very hard to read
4) Line 97/98, although a reference is given for the material parameters, perhaps the authors could provide a brief note explaining which type of steel (either a specific alloy, or a general classification) is being considered.
5) Line 146/147: What causes the localisation of stresses described by the authors?
6) Could the authors add a diagram or otherwise specify along which radial line the stresses are being shown?
7) The authors can also consider indicating the radius of the core wire in figure 3a.
8) Figure 3 and following figures: keeping the colours consistent for the different load cases would help in comparing the figures.
9) Figure 2: Consider rearranging the panels to form two columns: the left column showing the different von Mises distributions, and the right column showing the different max principal stress distributions. This would make it easier to compare
10) Figure 4: Again consider rearranging the panels to form two columns: Umag on the left, U3 on the right. Additionally, the scale of the U3 images should be adjusted to show meaningful differences, unless the authors intend to show that U3 is the same everywhere (which should then be highlighted in the text).
11) Figure 5: What is meant by 'distance'? Radial distance? Again, indicate along which radial line the graph is plotted, and consider indicating the core wire radius in the figure.
12) To make comparison easier, match the axis ranges of the two figures, and ensure the colours are consistent. What is meant by the additions _pt1 and _pt2 in the legend?
Since the point of the figure (if I understand the authors correctly) is to highlight the relative displacement of the wires for different friction conditions, the authors may wish to consider plotting all graphs in the same figure. If that would be confusing, perhaps a better solution is to have different sub-figures per maximum load, rather than per friction condition.
There are also some minor spelling issues / typos which can be fixed before final publication:
13) Due to the conventions regarding the spelling of German surnames, the correct capitalisation is 'von Mises', not 'Von Mises', i.e. with a lowercase v (unless it's the first word of a sentence)
14) Line 142: KN -> kN
15) Line 265: pikes -> peaks
16) Line 271: ahmad -> Ahmad
17) Line 277: the -> The
18) Line 295: Von mises -> von Mises
19) Figures 2, 4, and 5: Current images are blurry / contain compression or resizing artefacts. If possible, higher resolution / better quality versions should be included for final publication.
Reviewer 2 Report
This article is show the results of an interesting research about the behavior of Steel Wire under fretting fatigue conditions.
Minor changes could be made to improve the paper quality.
Legends in figures 2,3,4,5,6 and 9 should be improved. The text is to small. Position of nodes on outer wire are difficult to identify in figure 8.
Introduction can be improved, it should be cleared, better organized and enhance the latest advances in the field.
